# Anti-Adipogenic Effect of Theabrownin Is Mediated by Bile Acid Alternative Synthesis via Gut Microbiota Remodeling

**DOI:** 10.3390/metabo10110475

**Published:** 2020-11-23

**Authors:** Junliang Kuang, Xiaojiao Zheng, Fengjie Huang, Shouli Wang, Mengci Li, Mingliang Zhao, Chao Sang, Kun Ge, Yitao Li, Jiufeng Li, Cynthia Rajani, Xiaohui Ma, Shuiping Zhou, Aihua Zhao, Wei Jia

**Affiliations:** 1Center for Translational Medicine and Shanghai Key Laboratory of Diabetes Mellitus, Shanghai Jiao Tong University Affiliated Sixth People’s Hospital, Shanghai 200233, China; bernardk@sjtu.edu.cn (J.K.); joyzheng99@sjtu.edu.cn (X.Z.); huangfengjie1@outlook.com (F.H.); balihewangshouli@gmail.com (S.W.); limengci@sjtu.edu.cn (M.L.); echo_zml@sjtu.edu.cn (M.Z.); sang_chao@sjtu.edu.cn (C.S.); gk13127692129@outlook.com (K.G.); zhah@sjtu.edu.cn (A.Z.); 2School of Chinese Medicine, Hong Kong Baptist University, Kowloon Tong, Hong Kong, China; 20481373@life.hkbu.edu.hk (Y.L.); lijiufeng99@gmail.com (J.L.); 3University of Hawaii Cancer Center, University of Hawaii at Manoa, Honolulu, HI 96813, USA; crajani@cc.hawaii.edu; 4State Key Laboratory of Core Technology in Innovative Chinese Medicine, Tasly Pharmaceutical Co. Ltd., Tianjin 300410, China; maxiaohui@tasly.com (X.M.); zhousp@tasly.com (S.Z.)

**Keywords:** bile acids, gut microbiota, 7α-dehydroxylation, energy metabolism

## Abstract

Theabrownin is one of the most bioactive compounds in Pu-erh tea. Our previous study revealed that the hypocholesterolemic effect of theabrownin was mediated by the modulation of bile salt hydrolase (BSH)-enriched gut microbiota and bile acid metabolism. In this study, we demonstrated that theabrownin ameliorated high-fat-diet (HFD)-induced obesity by modifying gut microbiota, especially those with 7α-dehydroxylation on the species level, and these changed microbes were positively correlated with secondary bile acid (BA) metabolism. Thus, altered intestinal BAs resulted in shifting bile acid biosynthesis from the classic to the alternative pathway. This shift changed the BA pool by increasing non-12α-hydroxylated-BAs (non-12OH-BAs) and decreasing 12α-hydroxylated BAs (12OH-BAs), which improved energy metabolism in white and brown adipose tissue. This study showed that theabrownin was a potential therapeutic modality for obesity and other metabolic disorders via gut microbiota-driven bile acid alternative synthesis.

## 1. Introduction

Theabrownin is one of the most bioactive and abundant compounds in Pu-erh tea, a complex water-soluble polyphenolic substance produced with the fermentation of Pu-erh tea. The main constituents of theabrownin are polyphenols and products of oxidative polymerization of polyphenols with caffeine, proteins, sugars, and amino acids [1,2]. Our previous study [3] has well established that Pu-erh tea and theabrownin have cholesterol-lowering properties via reshaping gut microbiota and bile acid composition in mice and humans. Besides, we also observed the significant effects of theabrownin on triglyceride and energy metabolism involving energy expenditure from adipocytes, which have never been reported in our previous publication. Whether these effects involved theabrownin reshaping gut microbiota and bile acids and whether there existed different mechanisms remained elucidated.

Bile acids are the end products of cholesterol catabolism in the liver, playing a critical role in maintaining whole-body cholesterol homeostasis and regulating glucose and lipid metabolism as signaling molecules [4]. The two routes for BA synthesis are the classical pathway and the alternative pathway, mainly generating cholic acid (CA) and chenodeoxycholic acid (CDCA), respectively [5]. Actually, the ratio of CA to CDCA is determined by the expression level of CYP8B1, which transforms a di-hydroxylated BA to tri-hydroxylated BA [6]. Disturbance of the BA profile of elevated 12OH-BAs including CA, deoxycholic acid (DCA), and its conjugates has been found in obese patients with metabolic disorders [7]. Similarly, it has been reported that 12OH-BAs were significantly higher in T2D patients, and the ratio of 12OH to non-12OH BAs including CDCA, lithocholic acid (LCA), muricholic acids (MCAs) and its conjugates was positively correlated with key metabolic features including glucose and triglyceride (TG) levels [8,9]. The physiological implications of BA biosynthesis shift in various metabolic status remain to be explored. Lately, BAs have been recognized as signaling molecules involved in glucose, lipid, and energy metabolism through exerting either agonist or antagonist effects on the G protein-coupled BA receptor 1 (GPBAR1, TGR5) and the Farnesoid X receptor (FXR) [10]. Regarding the effects of BAs on energy metabolism, TGR5 was expressed in white [11] and brown adipocytes [12]. Besides, TGR5 as a critical regulator of energy expenditure, its activation induces the expression of iodothyronine-deiodinase type 2 (DIO2), converting inactive thyroxine (T4) into 3,5,3′-triiodothyronine T3, which stimulated the thyroid receptor to uncouple oxidative phosphorylation via mitochondrial uncoupling protein (UCP1), increasing thermogenesis and dissipating excess energy [13]. It was hypothesized that oral administration of CDCA to healthy humans increased energy expenditure via TGR5 activation in brown adipocytes [12]. Thus, manipulating TGR5 in adipocytes may be regarded as a promising target for the alleviation of obesity.

The gut microbiota is now considered an indispensable metabolic “organ” that facilitates the transformation of nutrients from food and produces countless metabolites to maintain a balanced host metabolism. Bile acids from the liver are often metabolized by the gut microbiota in the intestine, thus shaping the BA pool composition, extensively involved in the pathogenesis and progression of diseases. In the intestine, the conjugated bile acids are further metabolized by the gut microbiota through a series of deconjugation and dehydroxylation reactions, converting CA and CDCA to DCA and LCA, respectively [14]. Recently, research found amino acid conjugations of host bile acids by gut microbiota modifications, which has not been well characterized before, although their physiological roles remained to be established [15]. The condition known as gut dysbiosis often exhibits a changed composition of microbiota structure with an increased ratio of Firmicutes/Bacteroides [16], and this has been found in obesity, type 2 diabetes, and fatty liver diseases. Therefore, targeting microbiota may present a new approach for therapeutic intervention of these metabolic diseases.

Our study observed that host energy metabolism after theabrownin intervention has improved by examining the mRNA expression levels of UCP1 and peroxisome proliferator-activated receptor-gamma coactivator α (PGC1α) in adipose tissue. Furthermore, theabrownin remodeled gut microbiota structure in an HFD-induced obesity state by elevating the abundance of 7α-dehydroxylation-riched microbes, including *Clostridium scindens*, *Parabacteroides distasonis* which, in turn, resulted in increased levels of ursodeoxycholic acid (UDCA), CDCA, and LCA in the intestine. And the beneficial effects of theabrownin were in a gut microbiome-dependent manner through fecal microbiota transplant (FMT) validation. Additionally, after theabrownin administration, hepatic bile acid synthesis shifted from the classic pathway to predominantly the alternative pathway, resulting in a modified 12OH-BAs/non-12OH-BAs ratio in the liver and serum. These experiments revealed that the theabrownin-induced lipid-lowering effect could be partially attributed to BA-mediated energy expenditure in adipose tissue, which could have potential value for preventing and treating obesity.

## 2. Results

### 2.1. Metabolic Phenotypes and Host Energy Metabolism after Theabrownin Administration in HFD-Induced Obesity Mice Were Improved

To uncover the role of theabrownin in alleviating obesity, we established an obesity murine model in which male C57BL/6 J mice were fed with a high-fat-diet (HFD group) or high-fat-diet with theabrownin (HFD + TB group) for 8 weeks from 6 weeks of age. After 8 weeks of intervention, the mice body weights in the HFD group climbed progressively. In contrast, those in the HFD + TB group slightly increased, with significantly lower body weights on the 7-th and 8-th week than those in the HFD group (Figure 1A). Decreased liver weights indicated the amelioration of excessive hepatic lipid storage (Figure 1B) as confirmed by histological features (Figure 1C). In contrast with the HFD group, the HFD + TB group exhibited alleviation of diffuse hepatic steatosis (lipid accumulated as vacuoles with ballooning of hepatocytes) and adipocyte hypertrophy from epididymal white adipose tissue (eWAT)_(Figure 1C). Meanwhile, the HFD + TB group showed a lower triglyceride (TG) level in the liver (Figure 1D) and serum (Figure 1E) compared with the HFD group.

Manipulating energy harvest from the intestine and energy expenditure from adipose tissue have been promising approaches to combat obesity and related metabolic disorders. In our study, the mass ofeWAT as representative of visceral fat and subcutaneous white adipose tissue (sWAT) were reduced after theabrownin intervention (Figure 2A). But the mass of brown adipose tissue (BAT) remained unchanged between the two groups (Figure 2A). Investigation of the expression of WAT beiging- and lipolysis-related markers showed a marked upregulation of UCP-1, PGC1α, and elongase of very long chain fatty acids-3 (Elovl3) upon theabrownin treatment (Figure 2B). However, markers of lipolysis stayed statistically the same between the two groups (Figure 2B). Also, several markers of BAT energy metabolism were quantified. As depicted in Figure 2C, mRNA levels of UCP1, Elovl3, iodothyronine deiodinase 2 (Dio2) were upregulated, and Elovl6 was unchanged. To further decipher whether energy absorption from the intestine differed between the two groups, markers related to lipid absorption from the jejunum were assessed. Through the analysis of qPCR results, CD36 (a fatty acid translocase), long-chain fatty acid transport 4 (FATP4), and fatty acid-binding protein-1 (FABP1) had a downward trend but with no statistical significance (Figure 2D). The marker of energy metabolism, UCP1, was drastically increased in the BAT and WAT by theabrownin administration as shown by immunohistochemical staining (Figure 2E–H). Given the data above, there were no significant changes in adipocyte lipolysis and intestinal fat absorption; thus, there was reason to suspect that theabrownin exerted its lipid-lowering effect partly through promoting adipose energy consumption.

### 2.2. Theabrownin Reshaped the Gut Microbiome and BA Composition with Higher Non-12OH BAs

Our previous results based on 16S rRNA sequencing data have shown the structural remodeling of the gut microbiome in response to theabrownin. Here, we determined the effect of theabrownin on gut microbiota by whole shotgun metagenomic sequencing of intestinal content samples obtained from the ileum of HFD and HFD + TB groups at week 8. Results showed that the species richness was significantly higher in the HFD + TB group based on α diversity analysis (including Chao1 and Shannon index) (Figure 3A). UniFrac distance-based principal coordinate analysis (PCoA) displayed distinct clustering of intestinal microbe communities between the two groups, which implied the dissimilarity of gut microbiome after theabrownin intervention (Figure 3B). It has been reported that the gut microbiota in obese subjects possessed a higher Firmicutes to Bacteroides (F/B) ratio compared with lean controls [17]. Consistent with previous research, the gut microenvironment in the HFD group trended towards an increase in Firmicutes, and the F/B ratio declined dramatically after theabrownin intervention (Appendix A). Notably, at the species level, the top 15 gut microbes responsible for separating the two groups based on the VIP scores of partial least squares discriminant analysis (PLS-DA) contained *Akkmermansia muciniphila*, *Clostridium scindens*, *Streptococcus thermophilus*, *Enterobacter cloacae complex sp. 35734*, *Parabacteroides distasonis*, *Enterobacter asburiae*, *Bacteroides fragilis*, *Prevotella intermedia*, *Enterobacter cloacae*, *Prevotella dentalis*, *Staphylococcus capitis*, *Streptococcus intermedius*, *Blautia obeum*, and *Roseburia hominis* (Figure 3C). Fold changes based on microbial relative abundances between the two groups revealed that *Akkmermansia muciniphila*, *Clostridium scindens*, *Enterobacter cloacae complex sp. 35734*, *Parabacteroides distasonis* and *Enterobacter cloacae* sharply increased along with decreased relative abundances of other microbes (Figure 3C).

To further establish the correlation between gut microbes and BAs with theabrownin intervention, volcano plot of all intestinal BAs was first conducted. Results indicated that apart from considerably changed conjugated- and nonconjugated-BAs, which was consistent with our previous research [3], 12OH-BAs and non-12OH-BAs altered significantly between the two groups (Figure 3D). Spearman correlation analysis was performed between the BAs and the relative abundances of the differential bacteria species discussed above (Figure 3E). Generally, 12OH-BAs and non-12OH-BAs showed distinct trends of correlation with microbes. *Akkermansia muciniphila*, *Clostridium scindens*, and *Parabacteroides distasonis* positively correlated with non-12OH-BAs and negatively correlated with 12OH-BAs. Also, UDCA, LCA, ω-muricholic acid (ωMCA), and hyodeoxycholic acid (HDCA) all had a significantly positive correlation with *Clostridium scindens*, *Akkmermansia muciniphila*, and *Parabacteroides distasonis*. Our previous research demonstrated that colonization with *Clostridium scindens* enhanced the levels of tauroursodeoxycholic acid (TUDCA), taurochenodeoxycholic acid (TCDCA), and tauro-β-muricholic acid (TβMCA) in the liver, along with a particular emphasis on the levels of non-12OH-BAs [18]. *Parabacteroides distasonis*, which has a high tolerance for 20% bile salts and can generate LCA and UDCA as confirmed by in vitro and in vivo experiments, alleviated hyperglycemia and hepatic steatosis when colonized in HFD-induced obese mice [19]. Additionally, we have already shown that UDCA attenuated HFD-induced obesity via expansion of the non-12OH-BAs pool that resulted from a switch from the classic to the alternative pathway of BA synthesis [18]. These results implied that reshaped gut microbiota by theabrownin intervention was involved in the secondary BA metabolism with 7α-dehydroxylation, which resulted in the changed intestinal BA composition with a higher ratio of non-12OH to 12OH BAs.

To further investigate whether BA profiles altered in serum through theabrownin intervention, BAs from serum were analyzed and we observed that BA composition in serum and ileal contents displayed a high similarity with increased levels of non-12OH-BAs and decreased 12-OH-BAs in the HFD + TB group in contrast with the HFD group (Figure 3F). Serum BAs including UDCA, LCA, ωMCA, HDCA, TβMCA, were elevated along with lowering taurocholic acid (TCA) levels (Appendix A). Examination of the BA constitution in ileal contents revealed that LCA, HDCA, TCDCA, TUDCA exhibited increasing levels and deoxycholic acid (DCA), 23-nordeoxycholic acid (23-nor-DCA), taurodeoxycholic acid (TDCA), TCA showed a decreasing trend after theabrownin administration (Appendix A).

### 2.3. Theabrownin Exerted a Beneficial Effect on Metabolic Phenotypes in a Gut Microbiota-Dependent Manner

To investigate whether the gut microbiome mainly determined the beneficial metabolic effects of theabrownin, germ-free mice were transplanted with microbiota from mice fed with HFD or HFD supplemented with theabrownin (Figure 4A). After HFD feeding for 8 weeks, metabolic phenotypes showed a striking difference between the FMT-HFD and FMT-HFD + TB groups. Mice that had received the microbiota from the HFD + TB group exhibited a lower TG level than the mice with microbiota from the HFD group, consistent with results from donors (Figure 4B and Appendix A). Besides, the mass of eWAT, sWAT decreased in the FMT-HFD + TB group, although BAT mass stayed unchanged between the FMT-HFD and FMT-HFD + TB groups (Figure 4C). And qPCR analysis of white adipocyte beiging- and brown adipocyte energy expenditure-related markers showed a consistent trend with our observations presented above (Figure 4D). Notably, we observed a larger fraction of non-12OH-BAs in serum, liver, and ileal contents in the FMT-HFD + TB group (Figure 4E), which provided evidence that gut microbiota could regulate the conversion of 12OH- to non-12OH- BAs. Overall, there was reason to believe that the BA biosynthesis shift to the alternative pathway could be attributed to the microbial involvement in secondary BA metabolism, which produced elevated UDCA, LCA, and HDCA.

### 2.4. Changed Hepatic Bile Acid Synthetic Pathway beneath Altered 12OH-BAs/non-12OH-BAs in the HFD + TB Group

To determine the causes behind the shifted BA pool composition in serum and intestine, hepatic BA profiles were measured, and results were analyzed. Results showed that individual BAs in the liver differed between the two groups with findings of elevated non-12OH-BAs, including CDCA, TCDCA, TUDCA, tauro-α-muricholic acid (TαMCA) and TβMCA in the HFD + TB group along with decreased 12OH-BAs like TCA (Figure 5A). Notably, total hepatic BAs and conjugated and unconjugated BA levels were mostly unchanged between the two groups (Appendix A). Consistent with the overall trend for hepatic BAs, we found a decreased proportion of 12OH-BAs in the HFD + TB group (Figure 5B). To decipher the underlying molecular mechanism behind the altered 12OH-BAs/non-12OH-BAs ratio, we assessed the mRNA expression levels of BA biosynthesis-related enzymes in the livers of the two groups. Downregulated expression of CYP7A1 and drastically upregulated CYP7B1 were observed in the HFD + TB group (Appendix A), yet gene expression of Cyp8b1 and Cyp27a1 were basically unaffected by theabrownin administration. We further conducted western blot to validate the altered expression of these genes at the protein level. The results were in line with qPCR results, with increased expression of CYP7B1 in the HFD + TB group, almost a 2-fold change compared with the HFD group, and reduced levels of CYP7A1(Figure 5C). Given these data, we hypothesized that there was a switch in the BA synthetic pathways from the classic (CYP7A1 and CYP8B1-mediated) to the alternative pathway (CYP27A1 and CYP7B1-mediated) resulting from the administration of theabrownin to HFD-induced obese mice (Figure 5D).

### 2.5. The Effect of Selective BAs Intervention on Host Energy Metabolism in Mice

Based on the data presented above, the alteration of serum BAs after theabrownin intervention was shown as increased non-12OH-BAs (TCDCA, TUDCA) and decreased 12OH-BAs (TCA) (Figure 6A). To verify whether non-12OH- and 12OH-BAs had a different effect on host energy metabolism, we conducted an in vivo oral administration experiment using these representative BAs. Intervention with 50 mg/Kg/day TCDCA or TUDCA showed a shrinking mass of sWAT and eWAT, similar to the HFD + TB group. Still, administration with 50 mg/Kg/day TCA showed an increasing mass of sWAT and eWAT compared with HFD + TB group, similar to the HFD group (Figure 6B). qPCR analysis of WAT beiging- and BAT energy metabolism-related markers showed that only TCDCA could induce the expression of UCP1, PGC1α, Dio2 in WAT, and BAT (Figure 6C,D). TUDCA intervention caused a slight upward trend compared with HFD group controls (Figure 6C,D). However, TCA did not have an apparent influence on energy metabolism in adipose tissue (Figure 6C,D). Given the data available, theabrownin changed the BA pool composition by modification of the gut microbiota to produce higher levels of non-12OH-BAs, which, in turn, promoted adipocyte beiging and energy consumption.

## 3. Discussion

In our study, we found that the structure of the gut microbiome was remodeled after theabrownin intervention with an increased abundance of 7α-dehydroxylation-enriched microbes like *Clostridium scindens*, *Parabacteroides distasonis* and subsequently increased levels of the non-12OH-BAs including LCA and UDCA. These changed intestinal BAs resulted in a shift in hepatic BA production to the alternative biosynthesis pathway, enlarging the non-12-OH-BAs pool, including CDCA, LCA, and their conjugates. Elevated LCA and CDCA in circulation promoted energy metabolism including adipocyte beiging and energy expenditure. Notably, CYP7B1 activity was dominant in BA synthesis after theabrownin intervention, while the classic pathway was suppressed. Targeting BA synthesis regulation from the classic to the alternative pathway showed a potential value for preventing or treating metabolic diseases.

CYP8B1 activity determines the ratio of CA to CDCA. Some studies have revealed that depletion or downregulation of Cyp8b1 led to the improvement of host metabolic status. Cyp8b1^-/-^ mice which lack CA, displayed reduced fat absorption, increased GLP-1 release, and improved β-cell function in a BA-dependent manner [20]. Loss of Cyp8b1 resulted in lower amounts of 12OH-BAs, including CA, a highly efficient BA in mixed micelle formation for dietary fat and cholesterol absorption [21,22]. Thus, lack of CA will result in slowing lipid hydrolysis and allowing lipid access to the lower intestine, which slows gastric emptying and reduces food intake via the gut hormones, GLP1 and PYY mediated GPR-119 signaling pathway [23].

CYP7B1 is central to the alternative pathway of BA synthesis. We observed that CYP7B1 expression was dramatically upregulated after theabrownin administration. Many factors can influence the transcription and expression of CYP7B1 [24]. Some studies found that the MCAs/CA ratio was a marker of BA hydrophobicity and it was strongly reduced in TGR5-KO mice, which have lower expression of CYP27A1, the enzyme that initiates the alternative BA synthetic pathway. On the other hand, the BA pool from TGR5-overexpressing mice was less hydrophobic than for wild-type mice [25,26]. Using Cyp7b1^-/-^ mice, one study demonstrated that induction of hepatic Cyp7b1 was a primary driving force for a cold-induced increase in BA synthesis, which, in turn, stimulated energy expenditure [27]. There remain some questions regarding the mechanisms beneath changed BA composition and whether other unknown molecular switches can regulate bile acid synthesis.

Previous research has already verified that UDCA upregulated expression of CYP7B1, suppressed that of CYP8B1, and thus modified 12OH-BAs/non-12OH-BAs ratio [28]. Actually, the gut microbiota behaves as a gatekeeper by deciding how much transformation of CDCA to UDCA occurs in the gastrointestinal tract. Since both 7α/β-dehydroxylase and 7α/β-hydroxysteroid dehydrogenases (HSDH) had been found in *Clostridium scindens*, and the abundance of this organism is relatively high after theabrownin administration, epimerization from CDCA to UDCA happens more readily [29]. 7α/β-HSDH enzymes have a higher affinity for dihydroxy BAs (CDCA and 7-oxo-LCA) than for trihydroxy BAs (CA and 7-oxo-DCA) which further buttresses an argument that *Clostridium scindens* can biotransform CDCA to LCA effectively [29]. Another contributing bacterium, *Parabacteroides distasonis,* also possessed the ability to biotransform primary BAs to secondary BAs [19]. However, differences in the efficiency of CA and CDCA conversion to secondary BAs have not been explored. Therefore, in vitro experiments are necessary to validate and dissect the importance of the BA-gut microbiota interplay in the progression of metabolic diseases like obesity.

When energy intake surpasses the threshold that our body can tolerate, the delicate energy balance will be broken. Regulating lipid absorption from the intestine and energy expenditure from adipocytes are key targets in treating obesity. BA-induced activation of TGR5 has been shown to promote intracellular thyroid hormone activation to increase BAT energy expenditure [13]. Previously, CDCA was shown to induce mitochondrial uncoupling by activating TGR5 in human brown adipocytes [12]. Additionally, under cold exposure or in the HFD mouse model, BA triggered sWAT beiging and mitochondrial fission in a TGR5-dependent manner, alleviating obesity [30]. We observed that expression of UCP-1 was elevated in BAT after theabrownin intervention accompanied by increased expression of WAT beiging markers. Obviously, regulating the BA-TGR5 axis in adipose tissue has a strong potential for translation to the clinic to treat obesity.

There still exist many limitations to our study. Intestinal fatty acids should be measured, and calorimetry should be utilized to evaluate energy expenditure from feces. Additionally, in this experiment, we observed elevated levels of HDCA in serum and intestine, which has played an indispensable part in glucose metabolism from our latest research (unpublished data). Therefore, the impact of theabrownin on glucose homeostasis should be explored. Besides, intervention experiments using the driving bacteria (*Akkmermansia muciniphila*, *Clostridium scindens*, and *Parabacteroides distasonis*) should be performed to define the inter-relationships between microbiota and BAs further.

In summary, our research established that the gut microbiota facilitated the metabolism of primary to secondary BAs in the gastrointestinal tract after theabrownin intervention, leading to a shift in hepatic BA synthesis pathway and increased energy expenditure from adipose tissue (Figure 7). Modification of the BA-gut microbiome axis using a dietary supplement, theabrownin, has been shown to have the potential for alleviating obesity.

## 4. Materials and Methods

### 4.1. Theabrownin Extraction

Crude extraction: 350 g of Pu-erh tea (Menghai Tea Factory, TAETEA Group, Kunming, China, 7572) were milled into powder, suspended in a 10-fold volume of absolute ethanol, mixed for 12 h, and filtered under vacuum. The residue was extracted with a 10-fold volume of boiled distilled water, kept at 83 °C for 20 min with continuous stirring, and then filtered under vacuum. The extraction process was repeated three times, the extracts were combined and then vacuum evaporated to one-fifth of the total volume. The concentrated solution was then subjected to a series of liquid-liquid extraction processes, including equal volumes of chloroform, ethyl acetate, and n-butanol for 2, 3, and 4 times, respectively. The water layers were evaporated to one-quarter of their total volume and absolute ethanol added to a final proportion of 85% to precipitate the theabrownin crude extracts.

Purified extraction: The extracted crude theabrownin was further purified using a Sevage method. In general, the theabrownin samples were dissolved in distilled water and extracted with a chloroform/n-butanol mixture (5:1, v/v) repeatedly until no precipitated white turbidity was present at the liquid junction region. Absolute ethanol to a proportion of 85% was added to the water layer to precipitate the deproteinized theabrownin. The purified theabrownin was filtered and lyophilized for use.

### 4.2. Animal Study

All animal studies were performed following the rules dictated by national legislation and were approved by the Institutional Animal Care and Use Committee at the Center for Laboratory Animals, Shanghai Jiao Tong University Affiliated Sixth People’s Hospital (Shanghai, China) (committee approval code: DWSY2019-148).

In the theabrownin intervention study, 4-week-old mice were acclimated by placing them on a control chow diet ad libitum for one week and then randomly dividing them into two groups, eight mice per group: (1) high-fat diet group (HFD) (The high-fat diet contained 45% lipids, 19% proteins, and 36% carbohydrates.), and (2) HFD with 1.5 mg/mL theabrownin infusion group (HFD + theabrownin). The dosage of theabrownin was 225 mg/Kg per day and the intervention lasted for 8 weeks.

In the fecal microbiota transplantation study, the microbiota donors were mice treated with HFD or HFD supplemented with theabrownin for 8 weeks. Feces of the donors were collected and dispersed in sterile Ringer working buffer and the supernatant were mixed with skimmed milk for transplantation. Four-week-old germ-free male C57BL/6J mice were randomly divided into two groups (8 each group), housed in sterile plastic package isolators (each group for one isolator) and supplied with sterilized normal diet. After a 2-week acclimation, germ-free mice were oral gavaged with fecal suspension from mice with HFD or HFD supplemented with theabrownin. The oral gavage was repeated in the next two days to reinforce the microbiota transplantation. The transplanted mice were then fed with HFD for another 8 weeks. The body weight was recorded once a week and blood samples were collected at the end of the experiment for analysis of TG.

In the BA treatment study, 3-week-old mice were adapted with HFD for one week and subsequently divided into five groups, eight mice for each group: (1)vehicle fed HFD (HFD group), (2) HFD supplied with 1.5 mg/mL theabrownin infusion (HFD + theabrownin group), (3) HFD coupled with 50 mg/Kg body weight of TCA by gavage (HFD + TCA group), (4) HFD coupled with 50 mg/Kg body weight of TCDCA by gavage (HFD + TCDCA group), and (5) HFD coupled with 50 mg/Kg body weight of TUDCA by gavage (HFD + TUDCA group). The interventions were conducted for 8 weeks.

### 4.3. Biochemical Analysis

For the animal study, the levels of TG, TC in serum were measured using an automatic biochemical analyzer (Hitachi 7600, Tokyo, Japan). The liver TG content was detected using a triglyceride assay kit (A110-1-1, Jiancheng Bioengineering Institute, Nanjing, China).

### 4.4. Metagenomic Sequencing Analysis

The microbial genomic DNA samples were extracted using the DNeasyPowerSoil kit (QIAGEN, Valencia, CA, USA). The quantity and quality of extracted DNAs were measured using a NanoDrop ND-1000 spectrophotometer (Thermo Fisher Scientific, Waltham, MA, USA) and agarose gel electrophoresis, respectively. The quality-checked DNA sample was used to construct metagenome shotgun sequencing libraries by using the Illumina TruSeq Nano DNA LT Library Preparation Kit. Each library was sequenced by the Illumina HiSeq X-ten platform (Illumina, San Diego, CA, USA) with PE150 strategy at Personal Bio- technology Co., Ltd. (Shanghai, China). Raw sequencing reads were processed by FastQC (http://www.bioinformatics.babraham.ac.uk/projects/fastqc/) to conduct quality control. The sequencing adapters were removed from sequencing reads using Cutadapt (v1.2.1) (NBIS, Uppsala, Sweden). Low-quality reads, reads with ambiguous bases were removed using a sliding-window algorithm. The sequencing reads were aligned to the host genome and host genome reads were removed. Then quality-filtered reads were de novo assembled to construct the metagenome for each sample. Scaffolds/Scaftigs sequences were constructed by megahit (https://hku-bal.github.io/megabox/) of not less than 300 bp were selected for each sample.

### 4.5. Measurement of BAs

Bile acids in serum, liver, and intestine were measured according to methods previously established by our laboratory [31,32]. An aliquot of 25 μL of serum sample was mixed with 150 μL acetonitrile-methanol (8:2 v/v) containing 6 internal standards (IS) (D4-GCA, D4-GDCA, D4-CA, D4-UDCA, D4-LCA, and D4-GCDCA, 50 nM for each). All of the sample mixtures were allowed to stand at −20 °C for 10 min, and were then centrifuged at 13,000 rpm at 4 °C for 15 min. An aliquot of 150 μL of supernatant from each sample was transferred to another tube and then vacuum-dried. A total of 25 μL acetonitrile-methanol (8:2 v/v) was added. The sample was re-vortexed at 1500 rpm, 10 °C for 10 min, and 25 μL water containing 0.01% formic acid were added. The sample was vortexed again at 1500 rpm, 10 °C for 10 min, and then centrifuged at 13,000 rpm, 4 °C for 10 min. The supernatant from extraction was used for UPLC-MS analysis.

Raw data obtained from UPLC-MS were analyzed and quantified using Target-Lynx version 4.1 applications manager (Waters Corp., Milford, MA, USA).

### 4.6. Real-Time Quantitative PCR

Total RNA was isolated using Trizol reagent (Invitrogen, Carlsbad, CA, USA). The cDNA templates were obtained from 500 ng of purified RNA using iScript Reverse Transcription Supermix for RT-PCR (Bio-rad, Berkeley, CA, USA). Power Up SYBR Green PCR Master Mix (Applied Biosystems, Thermo Fisher Scientific) was used for quantitative RT-PCR, and assays were performed on an ABI Q7 PCR System (Applied Biosystems Instruments, Thermo Fisher Scientific). Targeted gene levels were normalized to housekeeping gene levels (GAPDH) and the results were analyzed using the ΔΔCT analysis method.

### 4.7. Western Blot Analysis

Liver samples were lysed with RIPA buffer (Beyotime Technology, Shanghai, China) containing 1 mM phenylmethylsulfonyl fluoride (PMSF) (Beyotime Technology) on ice. The concentration of total protein was quantified using a BCA Protein Assay Kit (Thermo, Waltham, MA, USA). A 5 µg/uL protein extract was added loading buffer (Beyotime Technology) and denatured by boiling at 100 °C for 10 min. Equal amounts of protein were electrophoresed on 10% SDS-page gels and transferred to a PVDF membrane. The membrane was blocked with 5% non-fat milk and then incubated with antibodies against CYP7A1 (1:1000, ab65596, RRID: AB_1566114, Abcam, Cambridge, MA, USA), CYP7B1 (1:1000, ab138497, RRID: AB_2828001, Abcam, Cambridge, MA, USA) and beta-actin (1:1000, #4970, RRID: AB_2223172, Cell Signalling Technology, Beverly, MA, USA) at 4 °C overnight. The membranes were washed three times by tris-buffered saline +Tween 20 (TBST) buffer and following a 2-h incubation with HRP conjugated secondary antibodies (1:2000, #7074, RRID: AB_2099233, Cell Signalling Technology). The blots were visualized using an ECL kit (Bio-Rad, CA, USA).

### 4.8. Immunohistochemistry Analysis

Liver, epididymal adipose tissue, and brown adipose tissue were embedded in paraffin and stained with routine hematoxylin and eosin (H&E). For the immunohistochemistry staining of UCP-1 in WAT and BAT, tissues were fixed in 4% paraformaldehyde and embedded in paraffin according to the manufacturer’s instructions. Then slides were deparaffinized, rehydrated, and stained with UCP-1 antibody (23673-1-AP, RRID: AB_2828003, Proteintech, Wuhan, China). Images were acquired through a digitalized microscope camera (Nikon, Tokyo, Japan) and further analyzed by ImageJ software.

### 4.9. Statistical Analysis

Results were presented as mean ± SEM. All the bar plots in this study were generated by GraphPad Prism 8.0 (GraphPad Software, La Jolla, CA, USA), and differential significance analysis using the Mann-Whitney U test and one-way ANOVA test was performed by SPSS 24.0 (IBM SPSS, Chicago, IL, USA) with criteria as * *p*-value < 0.05 and # *p*-value < 0.01. Correlations between BAs and microbiome relative abundances were performed using Spearman’s correlation analysis, visualized via R studio (RStudio, Boston, MA, USA). Partial least squares-discriminant analysis (PLS-DA) and Principal Coordinates Analysis (PCoA) were generated by R studio (RStudio).

## Figures and Tables

**Figure 1 metabolites-10-00475-f001:**
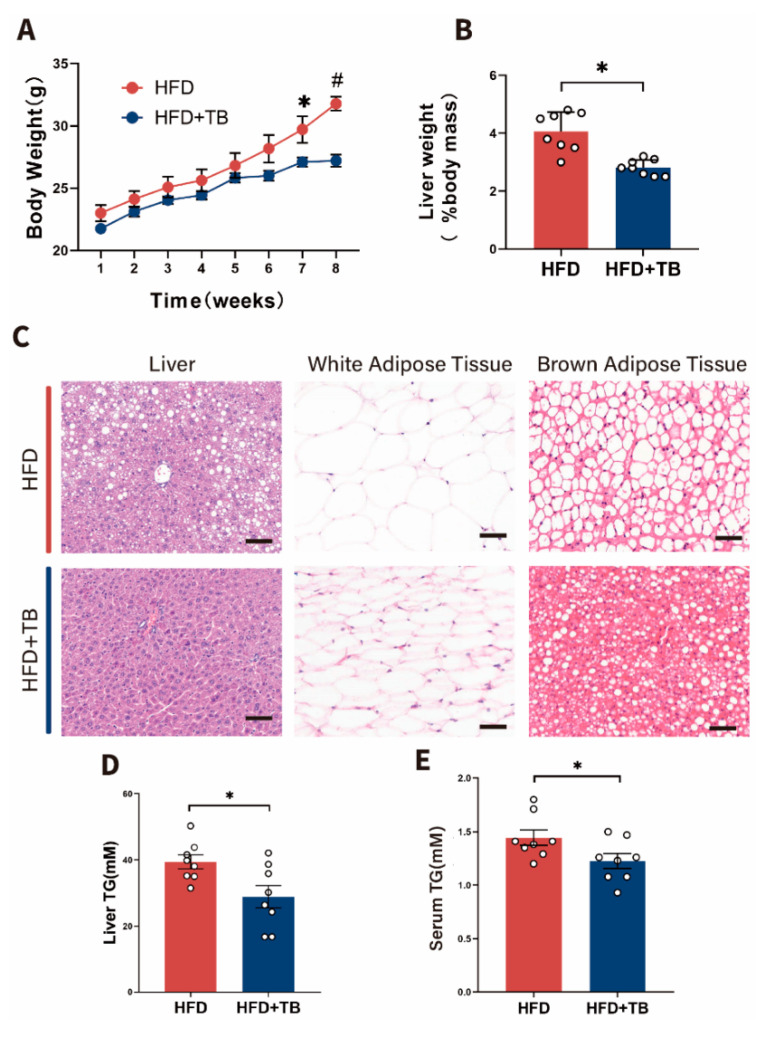
Metabolic phenotype changes after theabrownin intervention in HFD-induced obesity mice. (**A**) Body weights of mice intervened with theabrownin (HFD + TB group) in HFD-induced obesity mice (HFD group) for eight weeks. (n = 8 per group) (**B**) Percentage of liver weight to body weight between HFD and HFD + TB groups. (n = 8 per group) (**C**) Representative images of H&E staining of Liver, WAT, and BAT from HFD and HFD + TB groups, bars, 100 μm. WAT: white adipose tissue, BAT: brown adipose tissue. (**D**) Liver triglyceride content of mice between HFD and HFD + TB groups. (n = 8 per group) (**E**) Serum triglyceride content of mice between HFD and HFD + TB groups. (n = 8 per group) Data are presented as mean ± SEM. Differences between data were assessed using the Mann–Whitney U test, # *p* < 0.01, * *p* < 0.05.

**Figure 2 metabolites-10-00475-f002:**
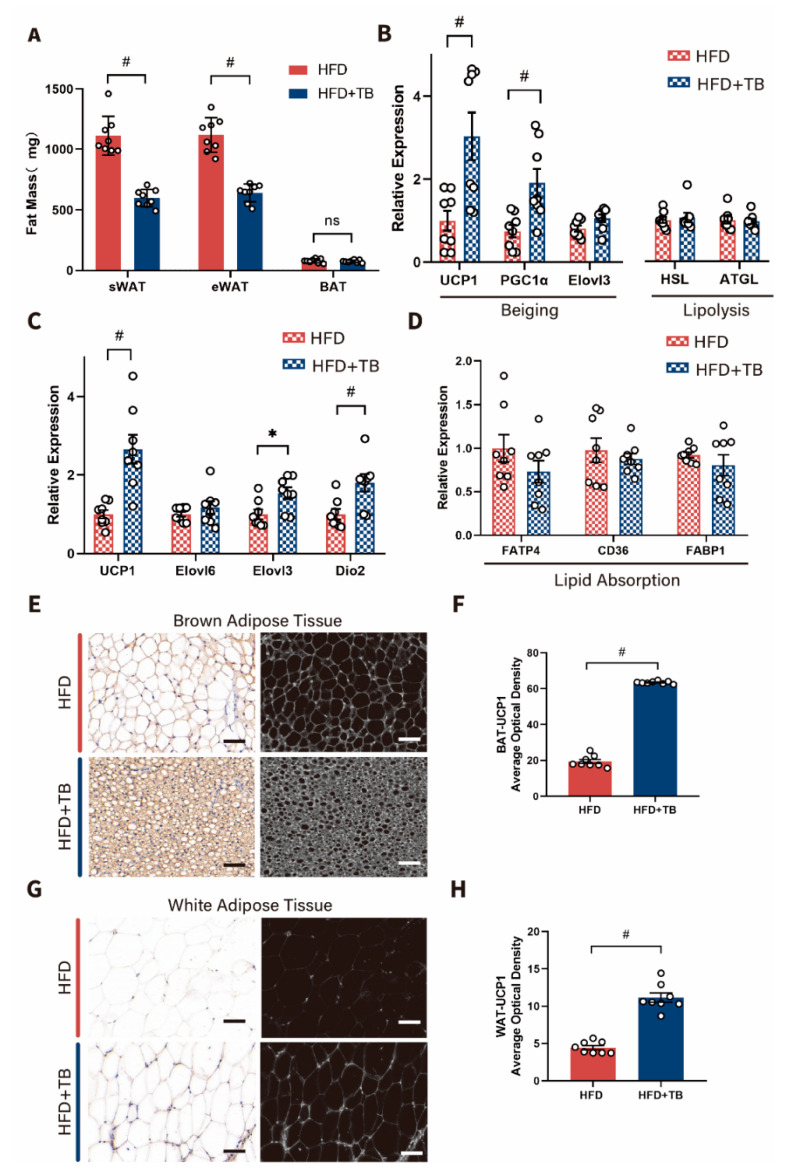
Increased energy expenditure in the HFD + TB group mice. (**A**) Subcutaneous white adipose tissue(sWAT), epididymal white adipose tissue (eWAT), brown adipose tissue (BAT) weight (mg) (n = 8 per group). (**B**) Relative mRNA levels of UCP1, PGC1α, Elovl3, HSL, and ATGL in white adipose tissue. (n = 8 per group). HSL: hormone-sensitive lipase, ATGL: adipose triglyceride lipase. (**C**) The mRNA expressions of UCP1, Elovl6, Elovl3, and Dio2 in brown adipose tissue (n = 8 per group). Elovl6: elongation of very long-chain fatty acids-6. (**D**) The mRNA expressions of FATP4, CD36, and FABP1 in the jejunum. (n = 8 per group) (**E**) Representative UCP1 immunostaining of BAT sections from HFD and HFD + TB groups. Scale bars, 100 µm. (**F**) Quantification of UCP1 immunostaining using an average optical density. (**G**) Representative UCP1 immunostaining of WAT sections from HFD and HFD + TB groups. Scale bars, 100 µm. (**H**) Quantification of UCP1 immunostaining using an average optical density. Data are presented as mean ± SEM. Differences between data were assessed using the Mann-Whitney U test, # *p* < 0.01, * *p* < 0.05.

**Figure 3 metabolites-10-00475-f003:**
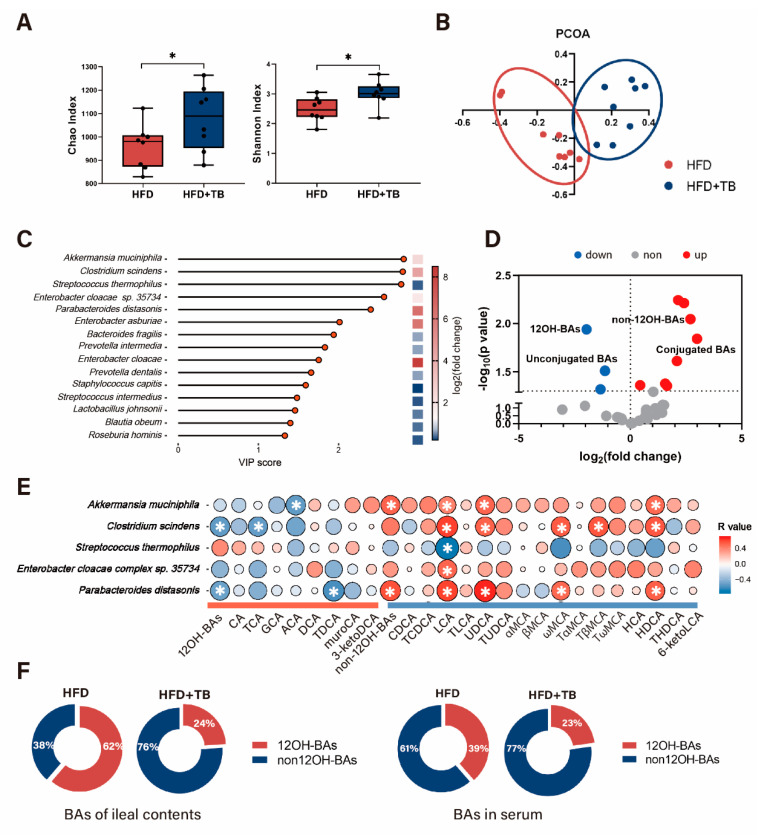
Remodeling of the gut microbiome and BAs constitution after theabrownin intervention in HFD-induced obese mice. (**A**) The α diversity analysis (Chao index and Shannon index) of metagenomic sequencing data. (n = 8 per group) (**B**) Principle coordinate analysis (PCoA) plot based on metagenomic sequencing data of mouse ileal contents in HFD and HFD + TB groups. (n = 8 per group) (**C**) VIP scores of PLS-DA analysis and fold changes (after log conversion) of HFD + TB to HFD group based on relative microbial abundance. (n = 8 per group) (**D**) Visualization of BAs in intestinal contents by volcano plots. Red dots represent relatively higher concentrations in the HFD + TB group compared with the HFD group; blue dots represent relatively lower concentrations in the HFD + TB group compared with the HFD group. (**E**) Spearman correlations of the relative BA abundance in ileal contents with the relative abundance of differential microbes based on VIP scores of PLS-DA analysis across the samples in the HFD and HFD + TB groups. (n = 8 per group) (Spearman’s correlation after the post hoc correction using the FDR method). (**F**) Altered percentage of non-12OH-BAs and 12OH-BAs of ileal contents and serum in HFD-induced obesity mice after theabrownin intervention for 8 weeks. Data are presented as mean ± SEM. Differences between data were assessed using the Mann-Whitney U test, * *p* < 0.05.

**Figure 4 metabolites-10-00475-f004:**
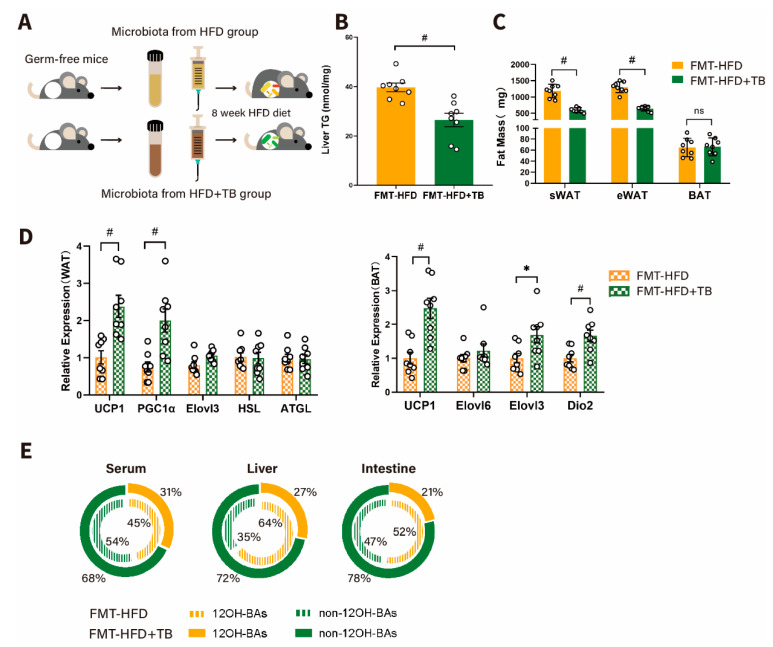
Reproducible beneficial phenotypes after performing a fecal microbiota transplant experiment. (**A**) Schematic diagram of the fecal microbiota transplantation experiment in germ-free mice. (n = 8 per group) (**B**) TG level in the liver of germ-free mice transplanted with microbiota from control (FMT+HFD group) and theabrownin (FMT-HFD + TB group) treated mice (n = 8 per group). (**C**) Subcutaneous white adipose tissue (sWAT), epididymal white adipose tissue (eWAT), brown adipose tissue (BAT) weight (mg) between the FMT-HFD and FMT-HFD + TB groups (n = 8 per group). (**D**) Relative mRNA levels of UCP1, PGC1α, Elovl3, HSL, and ATGL in white adipose tissue and UCP1, Elovl6, Elovl3, and Dio2 in brown adipose tissue (n = 8 per group). (**E**) Proportions of 12OH-BAs and non-12OH-BAs in serum, liver, and ileal contents for the FMT-HFD and FMT-HFD + TB groups. Data are presented as mean ± SEM. Differences between data were assessed using the Mann-Whitney U test, # *p* < 0.01, * *p* < 0.05.

**Figure 5 metabolites-10-00475-f005:**
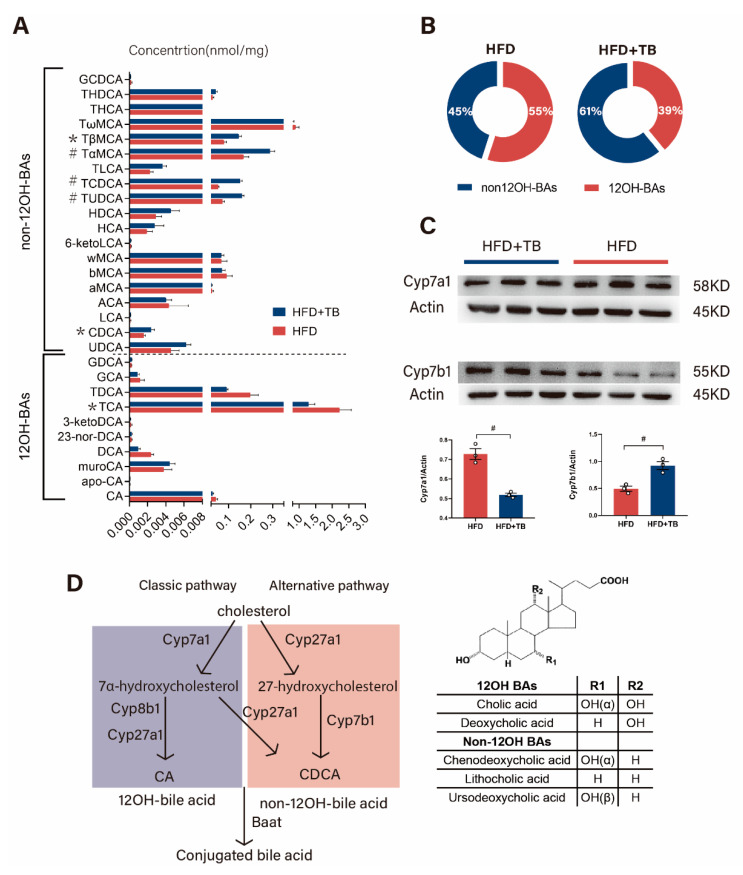
Altered hepatic bile acid profiles and bile acid biosynthesis by theabrownin intervention in HFD-induced obesity mice. (**A**) Altered hepatic BA profiles induced by oral administration of theabrownin for eight weeks in HFD-induced obesity mice (n = 8 per group). (**B**) Altered hepatic percentage of non-12OH BAs and 12-OH BAs in HFD-induced obesity mice after theabrownin intervention for eight weeks. (**C**) The expressions of CYP7A1 (n = 3 per group) and CYP7B1 (n = 3 per group) in the liver were detected by western blot. (**D**) Schematic diagram of the classical and alternative bile acid biosynthesis pathways. Data are presented as mean ± SEM. Differences between data were assessed using the Mann-Whitney U test. Statistical significance of BA data was corrected by the Benjamini-Hochberg method, # *p* < 0.01, * *p* < 0.05.

**Figure 6 metabolites-10-00475-f006:**
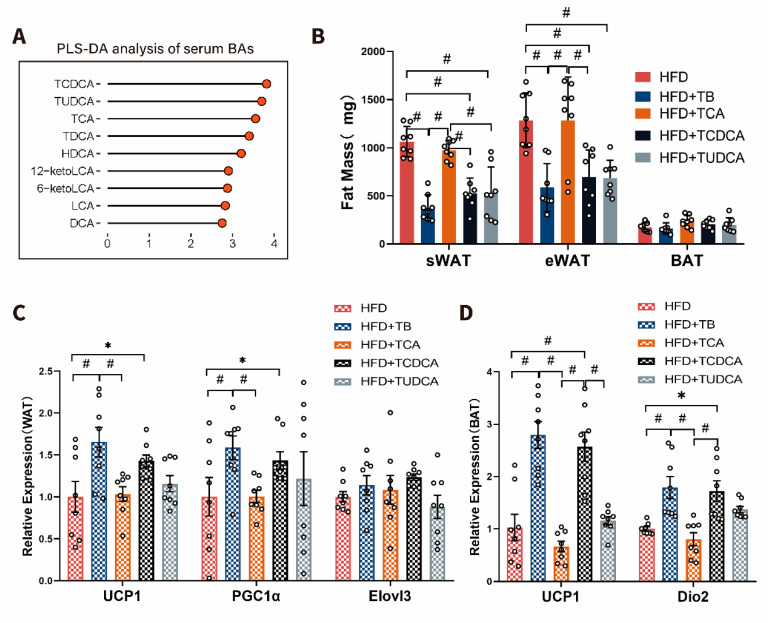
The impact of selective BAs intervention on energy metabolism in mice. (**A**) VIP scores of PLS-DA based on the serum BA profiles between the HFD and HFD + TB groups. (**B**) The sWAT, eWAT and BAT weights (mg) among the HFD, HFD + TB, HFD + TCA, HFD + TCDCA and HFD + TUDCA groups (n = 8 per group). (**C**) Relative mRNA levels of UCP1, PGC1α, and Elovl3 in white adipose tissue. (n = 8 per group). (**D**) The mRNA expressions of UCP1 and Dio2 in brown adipose tissue (n = 8 per group). Data are presented as mean ± SEM. Differences between data were assessed using the one-way ANOVA test, # *p* < 0.01, * *p* < 0.05.

**Figure 7 metabolites-10-00475-f007:**
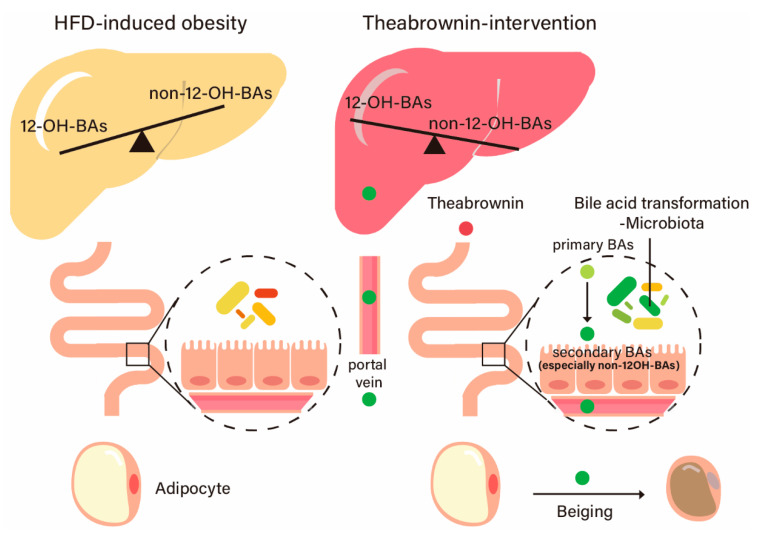
Schematic diagram of the mechanism of theabrownin’s action. Diet-induced obesity mice treated with theabrownin displayed a remodeled gut microbiota structure with the intensified activity of 7α-dehydroxylation, resulting in elevated secondary bile acids non-12OH-BAs like LCA, UDCA. Changed intestinal BAs returned to the liver through the portal vein and induced a shifting hepatic bile acid biosynthesis from the classic to the alternative pathway, leading to increased non-12OH-BAs and decreased 12-OH-BAs. Elevated levels of CDCA and LCA in circulation induced TGR5-dependent adipocyte beiging and energy expenditure to alleviate obesity.

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
