# Peer review of "Anti-Adipogenic Effect of Theabrownin Is Mediated by Bile Acid Alternative Synthesis via Gut Microbiota Remodeling"

_metabolites, 2020, doi:10.3390/metabo10110475_

Round 1
Reviewer 1 Report
Kuang et al. report the effect of theabrownin, a component of Pu-erh tea, in the context of lipid-lowering properties. The authors investigate changes in the microbiome composition and bile acids. They also investigate differences in white and brown adipose tissue (not mentioned in the abstract). This interdisciplinary study is of high interest for readers interested in the impact of the microbiome on the host and the metabolic link to diet. My enthusiasm drastically reduced when I found a huge overlap with a previous publication by the same authors (Nat. Comm. 2019). Reporting the same data is not appropriate with the high standards of Metabolites (and any other journal). While the authors report new results, the paper can only be rejected in the current form as the author must resubmit a completely rewritten manuscript with removal and clear distinction to their previous work. In most instances, the first paper has not been cited at the most important parts. This must be made absolutely different by the authors as it should not be the task of a reviewer to find the similarities.
Identical data between both papers (with slightly altered presentation or rewording):
No) Metabolites submission = Nat. comm. 2019
1) Fig. 1A = Suppl Fig. 10a
2) Fig. 7B = Suppl Fig. 10b
3) The shannon and chao index were already calculated in the first paper and are presented in here.
4) Figure 3A = Figure 5C
5) Fig. 4C = Fig. 7h
6) Fig. 4E = part of Fig. 8b
7) Fig. 4D - no novelty.
8) The experimental parts are nearly identical and thus the samples result from the same study and LC-MS analysis.
9) Most of the discussion is similar as well (BAs and CYP enzymes)
Minor:
The formatting of the manuscript is off. The authors have too many double-spaces and lack of spaces and references in the middle of sentences that should be corrected.
Reviewer 2 Report
This study shows that the gut microbiome composition is changed by theabrownin resulting in increased levels of the BAs, LCA and UDCA. Theabrownin induced a shift to the alternative hepatic BA biosynthesis pathway, which increased the pool of non-12-OH-BAs, including CDCA, LCA and their conjugates. Interestingly, adipocyte beiging and energy expenditure was stimulated by the elevated levels of LCA and CDCA. CYP7B1 activity became dominant in BA synthesis after theabrownin intervention, while the classic BA biosynthesis pathway was suppressed.
The authors propose that the effect of theabrownin on the adipose tissue "could have potential value for the prevention and treatment of obesity".
This is a well-designed study and the results obtained are interesting.
There is one major comment (1) and more minor comments (2-10), which show, however, that the proofreading of the manuscript has not been done carefully, which is a bit surprising given the number of authors.
Comments:
1) Although this study is different from a recently published study in Nature Communications by in part the same authors (Huang et al. Nat Commun. 2019 Oct 31;10(1):4971. doi: 10.1038/s41467-019-12896-x), it would help the reader if the differences between the two studies and how the studies are complementary is better explained.
2) Introduction – line 83. BSH is for Bile Salt Hydrolase. Please clarify.
3) Results – lines 107 to 110. The results of Fig. 1A are not well described. It is not correct to say that the weight “HFD+TB group tended to be stable” when there is an increase of 3-4 grams, which is quite significant in mice.
4) Results – line 191. The authors say that the “BA profiles in the serum, liver, and ileal contents” were analyzed. The liver results are not presented.
5) Results - Fig 4C. Please indicate what is HFD and HFD + TB.
6) Results – line 251 and Fig. 5C. There is no marked upregulation of Elovl3 upon theabrownin treatment. The difference is not significant.
7) Results – lines 259-261.There is a problem with the panel identification of Fig. 5. 5C in the text corresponds to 5D in the figure and 5D-G in the text to 5E-H in the figure. Correct? Fig. 5H labelling: ordinate WAT not AAT.
8) Results – line 274. What is meant by “previous data”. If published a reference should be given.
9) Results – lines 277-279. The sentence is not clear. “… increasing mass of sWAT and eWAT …”; increase compared to what?
10) Discussion – first sentence (lines 325-327). It is not said which condition induced a change in the microbiome
Reviewer 3 Report
In this manuscript the authors describe the effect of theabrownin (Tb) from Pu-erh tea on the gut microbiome, bile acid metabolism and lipid profiles of mice. The manuscript uses mass spectrometry and microbiome sequencing methods to measure the effects of many different bile acids and microbes, respectively, when adminstered theabrownin and high fat diet. Remarkably, Tb lowers the body weight of HFD fed mice and this is also seen as a reduction in liver adipose tissue white and brown fat. The authors then go on to demonstrate the effect of Tb on the gut microbiota and bile acid profiles, ultimately showing that the effect of theabrownin is mediated through a change in bile acid synthesis from the classical to alternative pathway. Overall, this manuscript is a neatly performed complete study of which I applaud the study design and analysis. Their findings are interesting and likely will be well received by those in the metabolomics community, but also those in obesity and the microbiome fields.
Comments and Critques.
Firstly, what is theabrownin and Pu-erh tea? There is no structural information about this compound and little background information provided. Throughout my review I was curious about this compound and remained so after reading the paper.
I have problems with the final sentence of the abstract. There data does not support such a broad claim. More accurately, it supports that Tb may be a therapy for obesity. This should be removed.
The introduction is quite long and could be reduced to focus more on the tea and theabrownin. The intro is also missing an important recent study that found another mechanism of microbial modification of bile acids through amino acid conjugation https://www.nature.com/articles/s41586-020-2047-9. The authors describe the actions of the microbiota on host bile acids, but leave out this recent finding.
Is the statement on line 140 incorrect? The authors describe the F/B ratio as elevated during obesity then say that Tb drastically increased it. This seems backwards.
It is not apparent from the figure or methods that the bile acid comparisons between HFD and HFD+TB have been corrected for multiple comparisons. There are 24 bile acids being tested in this figure creating a high likelyhood of false discovery. The effect size of the treatment is small in many cases, thus, FDR correction or better description of how this was done is needed. This is especially true in the liver data from Fig. 4 making these findings somewhat suspect without FDR correction.
Bile acid structure is upside down compared to the other structure in Fig. 4.
Reference to Fig. 5c on line 259 should be 5D.
Line 260. I don't consider a 3-fold increase to be drastic.
Round 2
Reviewer 1 Report
The authors have addressed all of my concerns and clarified differences between their previous study and this manuscript. I recommend publication of this revised and highly improved manuscript in Metabolites.
With these changes, this manuscript is a strong contribution. I am just surprised that the authors mention their previous manuscript by name in the abstract. With the changes made and updated figures it should be mentioned in the manuscript but not in the abstract. I recommend rewriting the abstract before publication.
Reviewer 2 Report
The revised version of the manuscript has been improved compared to the original version.
I would propose that the explanation of how this work relates to the previous work of this laboratory published in Nature Communications is removed from the Abstract and is given in the Introduction.
Reviewer 3 Report
The authors have satisfied my critiques, however, throughout the new version of the paper there are grammatical errors. I suggest extensive text editing before submitting a final version.
Author Response
Please see the attachment.

This manuscript is a resubmission of an earlier submission. The following is a list of the peer review reports and author responses from that submission.